# Brain–Computer Interface Speller Based on Steady-State Visual Evoked Potential: A Review Focusing on the Stimulus Paradigm and Performance

**DOI:** 10.3390/brainsci11040450

**Published:** 2021-04-01

**Authors:** Minglun Li, Dianning He, Chen Li, Shouliang Qi

**Affiliations:** 1College of Medicine and Biological Information Engineering, Northeastern University, Shenyang 110169, China; 1801223@stu.neu.edu.cn (M.L.); hedn@bmie.neu.edu.cn (D.H.); lichen@bmie.neu.edu.cn (C.L.); 2Engineering Research Center of Medical Imaging and Intelligent Analysis, Ministry of Education, Northeastern University, Shenyang 110169, China; 3Key Laboratory of Intelligent Computing in Medical Image, Ministry of Education, Northeastern University, Shenyang 110169, China

**Keywords:** brain–computer interface (BCI), speller, steady-state visual evoked potential (SSVEP), hybrid, paradigm, triggering method

## Abstract

The steady-state visual evoked potential (SSVEP), measured by the electroencephalograph (EEG), has high rates of information transfer and signal-to-noise ratio, and has been used to construct brain–computer interface (BCI) spellers. In BCI spellers, the targets of alphanumeric characters are assigned different visual stimuli and the fixation of each target generates a unique SSVEP. Matching the SSVEP to the stimulus allows users to select target letters and numbers. Many BCI spellers that harness the SSVEP have been proposed over the past two decades. Various paradigms of visual stimuli, including the procedure of target selection, layout of targets, stimulus encoding, and the combination with other triggering methods are used and considered to influence on the BCI speller performance significantly. This paper reviews these stimulus paradigms and analyzes factors influencing their performance. The fundamentals of BCI spellers are first briefly described. SSVEP-based BCI spellers, where only the SSVEP is used, are classified by stimulus paradigms and described in chronological order. Furthermore, hybrid spellers that involve the use of the SSVEP are presented in parallel. Factors influencing the performance and visual fatigue of BCI spellers are provided. Finally, prevailing challenges and prospective research directions are discussed to promote the development of BCI spellers.

## 1. Introduction

The brain–computer interface (BCI) is a communication system that allows humans to send messages and commands to the outside world without depending on peripheral nerves and muscles [1]. The BCI can be used for control and communication such that it provides a mode of communication for patients with motor neuron diseases (MNDs) such as amyotropic lateral sclerosis (ALS) and locked-in syndrome (LIS), to significantly improve their quality of life. Currently available devices for the BCI based on the electroencephalograph (EEG) are the most widely used because EEG signals can be easily collected on the subject’s scalp.

The BCI speller is a typical visual application of the BCI, and was among the earliest implementations of the concept. The first BCI speller was proposed by Farewell and Donchin [2]. It was based on event-related potential (ERP) P300, and featured a row and column (RC) paradigm. The potential of the ERP P300 in EEG signals is positive, and occurs at about 300 ms, with differences in the specific time of its occurrence, after stimulus caused by an event with a small probability. This potential is prominent in the central partial region of the brain. For a few years after its development, most BCI spellers that were proposed were based on the P300. In addition to the visual P300, the auditory P300 [3] and tactile P300 [4] were added to the design of the BCI. A problem with the P300 speller is its slow speed. As every output needs to traverse all stimuli of a holonomic trial, users need to wait for a long time. In addition, the amplitude of the potential of the P300 decreases with increasing use time and reduces its accuracy of classification [5].

The steady-state visual evoked potential (SSVEP) was proposed to design a high-speed BCI speller to solve the above problems. The signals of the SSVEP is part of the visual evoked potential (VEP), which is a kind of co-frequency and harmonic response caused by a stimulus at a specific frequency. It is usually evoked in a fixed and independent position of the occipital region of the brain and is measured by EEG equipment. The principle of the SSVEP-based BCI speller is to present a series of visual stimuli at specific frequencies to users and detect the SSVEP evoked by them through frequency domain analysis to find the user’s target. Other than that, SSVEP is also applied to BCIs for control, such as control wheelchairs [6] and robotic arms [7].

The advantage of the SSVEP-based BCI speller is that it does not require calibration or training for its users, and does not rely heavily on a large amount of training data. However, the frequencies that can be used to evoke the SSVEP are limited by the refresh frequency of the screen. Another problem is that the SSVEP-based BCI speller induces visual fatigue in the user, where such fatigue is not severe but cannot be eliminated [8]. A performance comparison between the SSVEP-based BCI speller and the P300-based speller has yielded differences between LIS patients and healthy subjects [9,10].

To improve the performance of BCI spellers, a number of spellers based on other triggering methods other than that mentioned above have been proposed. These methods include the motion-onset VEP (m-VEP) [11,12,13], code VEP (c-VEP) [14], and motor imagery (MI) [5,15,16,17]. The m-VEP-based BCI speller is also known as the N200 speller and is characterized by improved user experience, but is slow [13]. The c-VEP speller delivers better performance than the SSVEP speller but is not as user friendly [18]. BCI spellers based on MI spellers allow users to select the desired target through the imagination of the movement of limbs which induces a sensorimotor rhythm (SMR), such as by moving a cursor [5], and are independent of external stimuli. SMR is widely used in BCIs for control, such as being used for continuous control of the robotic device [19], and achieves good performance. Eye-tracking (ET) is also often used as the method to trigger visual speller applications, although it does not belong to the category of BCI [20]. The ET speller uses a camera to monitor the users’ eye movements to determine the target of their gaze.

BCI spellers developed using the SSVEP have received considerable research attention, and are being developed rapidly. A comprehensive review of BCI spellers has been provided in Reference [21], and reviews of specific classification algorithms, data analytics, and language models can be found in Reference [22,23,24], respectively. The paradigm of visual stimuli, including the procedure of target selection, layout of targets, manner of encoding of the stimuli (i.e., frequency and phase), and their combination with other triggering methods has an important influence on the performance of the BCI speller. However, no review of this work has been provided in the literature to date.

This review focuses on the stimulus paradigm and performance of BCI spellers that use the SSEVP. The prevalent SSVEP speller is classified and discussed according to the stimulus paradigm, and methods to optimize the performance of the SSVEP speller are discussed. The main purpose is to update the reader on progress in the area, summarize the trends and available challenges, and present some directions for future research in the area.

The remainder of this paper is organized as follows: Section 2 introduces the basic contents of the SSVEP speller, including the system architecture and process, and methods of SSVEP acquisition and detection as well as the performance evaluation. Section 3 presents several representative paradigms of SSVEP-based BCI spellers. Section 4 illustrates the paradigms of the SSVEP combined with hybrid BCI spellers, and Section 5 details factors influencing their performance and the problem of visual fatigue in the user. The trends in the area, outstanding challenges, and directions for future research are given in Section 6. Section 7 summarizes the conclusions of this study.

## 2. BCI Speller Based on SSVEP

### 2.1. System Architecture and Process

The architecture and process of an SSVEP-based BCI speller is shown in Figure 1. The SSVEP speller requires four devices: those for stimulus presentation, data acquisition, data processing and feedback control. In applications, stimulus presentation, data processing and feedback control can be carried out on the same computer, but we consider them separately to facilitate discussion according to their process and function. This architecture is similar to that in the BCI2000 system proposed in Reference [25].

The stimulus presentation device presents stimuli to the users that can evoke the SSVEP. The data acquisition device records and digitizes the user’s scalp EEG that contains SSVEP so that it can be processed and analyzed by the data processing device. The data processing device converts digitized EEG signals into commands that can control the system. These commands are sent to the feedback control device, which performs functions corresponding to the commands received from the data processing device and gives feedback to the user. It presents the non-visual feedback (such as sound) directly through other devices and visual feedback on the graphical user interface (GUI) presented on the stimulus presentation device to the user.

Users need to look at the target presented on the GUI that can evoke the SSVEP signal when they use the speller. During this period, the user’s scalp EEG is recorded by the data acquisition device and sent to the data processing device in digital form. These EEG data need to be preprocessed first, where this generally includes artifact removal, baseline correction, filtering, and data segmentation. The preprocessed data are then used for SSVEP detection. The results of detection are classified by a classifier to determine whether the user was staring at the target and identify the one at which they were staring. The results of classification are translated into commands according to the system design and sent to the feedback control device. The controller in the feedback control device changes the content presented on the GUI according to the received commands and presents new stimuli or visual feedback to the users, such as displaying the selected characters in a text box so that they can proceed to the next operation. Furthermore, this device can also control the system states, such as pause, continue, or terminate. Other feedback such as sound will also be presented by other devices which are not presented in this figure. Feedback methods are proved to be a factor affecting performance [26,27], which will be mentioned in later sections.

### 2.2. SSVEP Acquisition

The SSVEP usually appears in the occipital region of the brain [28]. Most studies have used four to 11 electrode channels in the occipital region for data acquisition. Some studies have shown that single-channel acquisition using two electrodes is also feasible for SSVEP detection [29], whereas others have shown that SSVEP signals can also be collected from hairless regions of the body (including the neck, behind the ears, and face), but the signal-to-noise ratio (SNR) in this case is lower than that extracted from the occipital region [30].

### 2.3. SSVEP Detection Methods

The principle of SSVEP detection is to find the crest of the wave generated by an induced frequency and its harmonics through a frequency domain analysis of preprocessed EEG signals. The first method of detection used to this end was power spectrum density analysis (PSDA) as represented by a fast Fourier transform (FFT) [31,32]. With the subsequent introduction of new algorithms, PSDA has been shown to be inferior to the spatial filtering method, including the common spatial pattern (CSP) [33], minimum energy combination (MEC) [34], and canonical correlation analysis (CCA) [35]. CCA has attracted the most attention among these methods because of its high robustness and efficiency. It also outperforms the combination of linear discriminant analysis (LDA) and the support vector machine (SVM) in the SSVEP detection of the SSVEP speller [36]. Algorithms for SSVEP detection proposed in the same period as the spatial filtering method include the multivariate synchronization index (MSI) [37], common feature analysis (CFA) [38], and likelihood ratio test (LRT) [39], and have delivered good performance.

In later research, many improved algorithms based on CCA have been proposed. Reference [40] compared eight methods of SSVEP detection based on CCA: (1) the standard CCA [35], (2) cluster analysis of CCA coefficients (CACC) [41], (3) phase-constrained CCA (PCCA) [42], (4) multi-way CCA (MwayCCA) [43], (5) L1-reguralized multi-way CCA (L1-MCCA) [44], (6) multi-set CCA (MsetCCA) [45], (7) individual template-based CCA (IT-CCA) [46], and (8) a combination of the standard CCA and IT-CCA [47].The authors concluded that the latter achieves the best performance. Chen et al. proposed a method to incorporate components of the fundamental and harmonic frequencies to improve the detection of SSVEPs, named filter bank canonical correlation analysis (FBCCA) [48]. FBCCA was found to be superior to the standard CCA in an experiment on 10 subjects, and was long considered to be the best method for SSVEP detection. Reference [49] proposed a hybrid method that combines SSVEP data and FBCCA, named the ensemble/extended CCA, that recorded the best performance of available methods at the time.

In recent studies, task-related component analysis (TRCA) has been shown to be superior to the extended CCA in terms of accuracy and speed [50]. In addition, maximal-stage-locking value and minimal-distance spatial filter bank (MP and MD) have been proposed to optimize FBCCA, and can outperform the extended CCA [51]. In addition, deep learning methods have been used for SSVEP detection [29,52]. PodNet (a novel deep convolutional neural networks, DCNN) has been shown to be superior to FBCCA while using fewer channels [52].

Several datasets can be used for offline research, such as a 40-class SSVEP dataset recorded from 35 subjects [53] and a 12-class SSVEP dataset recorded from 10 subjects [40]. The frequencies as well as the phases are different in the dataset proposed in Reference [53]. Nakanishi et al. proposed a method of transfer learning across devices and montages [54,55] that significantly improves the availability of these datasets.

### 2.4. Performance Evaluation

Two important parameters are used to evaluate the performance of the BCI speller: accuracy and information transfer rate (ITR).

In most studies, accuracy has been defined as the ratio of the number of correct target selections to the number of total selections attempted by the system. It can be calculated as follows:(1)$$\mathrm{Acc}=\frac{X_{1}}{X}\times 100\%$$

In Equation (1), Acc represents accuracy, *X_1_* represents the number of correct inputs, and X represents the total number of inputs. The accuracy value can be used to evaluate the performance of the classification algorithm of a speller, which is an important parameter in applications. A higher accuracy can reduce the number of repeated user inputs to improve the efficiency of the system as well as user experience.

Although accuracy is an important index, the speller should also consider the speed of input of the system in applications. Wolpaw introduced the ITR to the evaluation of BCI systems [56] as a parameter for performance evaluation. The ITR (in bits per minute, bpm) can be written as follows:(2)$$\mathrm{ITR}=\frac{B}{T}$$
where *B* is defined as the amount of information transmitted in each round of experiments, and is written as
(3)$$B=\log_{2}N+P\log_{2}P+\left(1-P\right)\log_{2}\frac{1-P}{N-1}$$
where *N* is the number of categories in the system that can output commands and P is the probability of correctly selecting the target option. *T* is the time needed for each round of experiments. In different studies, researchers have used different definitions of *T*. Some did not consider the time taken by the subjects to select characters and the interval between rounds of selection, while others have calculated these. The former is better suited to assessing the algorithm while the latter focuses on system performance in applications.

There are three factors that affect the ITR: accuracy, the number of targets, and the time required. Some researchers have used Nykopp’s ITR [57] to evaluate the performance of the results of research, but this is not typical. Others have used their own standards to evaluate performance, such as the practical ITR (PITR) [58] and information gain rate (IGR) [59].

## 3. SSVEP-Based BCI Spellers

In early studies on the SSVEP-based BCI speller, most researchers used only a small number of low-frequency stimuli as they were limited by the frequency of refresh of the stimulus presentation devices. Hardware to present external stimuli independently of the computer monitor and the frequency–phase hybrid coding paradigm were subsequently proposed to enable the speller to present a large number of targets at the same time. The one-stage paradigm was applied to the design of the SSVEP speller. With the development of high-frequency stimulation technology, high-frequency stimuli were used in the SSVEP speller to improve the SSVEP SNR and reduce user fatigue. We subdivide the SSVEP speller according to paradigms and review each in chronological order, as shown in Figure 2.

### 3.1. Bremen Speller

The Bremen speller is a well-known paradigm of the SSVEP speller that presents all targets on the screen and uses five boxes to control the cursor. The five boxes flicker at different frequencies to evoke different SSSVEP responses, as shown in Figure 3. The user selects a target character by moving the cursor from one to five times. The relevant study achieved an average accuracy of 92.84%, and average ITRs of 22.6 bpm at the command level and 17.4 bpm at the speller level [60].

In a subsequent study, Volosyak et al. improved the standard Bremen speller by adding a dictionary for spelling prediction [61]. Based on the standard Bremen speller, a “Go” box was added to the user interface. Compared with the standard Bremen speller, this had higher ITRs of 32.71 bpm at the command level and 29.98 bpm at the speller level. In Reference [62], average ITRs of the Bremen speller of 61.70 bpm and 92.8 bpm were recorded for the fastest subject after improvement, where this speller had the highest ITRs at the time. However, in subsequent studies, multi-stage SSVEP spellers were shown to provide a more efficient paradigm for users [63].

### 3.2. Multi-Stage SSVEP Speller

The multi-stage SSVEP speller has different paradigm designs, but they all have the following characteristics: the user interface contains two or more stages and the number of targets in each stage is relatively small. Compared with one-stage SSVEP spellers, such spellers have the advantage of reducing the encoding of the frequency. However, this paradigm also reduces the efficiency of input of the user because they can select only the target character in the last stage. Compared with the Bremen speller, its performance is stabler because the number of operations is certain (the number of operations is equal to the number of layers in general), and it can realize the “what you see is what you choose” design.

The SSVEP speller proposed in Reference [64] is considered to be a classic multi-stage SSVEP speller that consists of three stages with five commands per stage: three commands to choose a letter, a “Previous Action” command to cancel the previous command, and a “Delete Character” command to delete the last written character, as shown in Figure 4. Users can find the characters they want by using one command in each of the three layers [64].

Saboor et al. proposed a three-stage SSVEP web speller that has four commands per stage and 27 character targets, and used the MEC to detect the SSVEP. The average accuracy and ITR of this study were 92.25% and 37.62 bpm, respectively [65]. In later research, spelling prediction based on the Leipzig Corpora Collection was added to this web speller to yield an average accuracy of 92.5% and an ITR of 18.8 bpm [66].

Nguyen et al. proposed a three-stage SSVEP speller with five commands per layer and 57 character targets. A 1D CNN was used to detect the SSVEP in single-channel EEG data. The average accuracy and ITR hence obtained were 97.38% and 48.99 bpm, respectively [29].

Other studies have proposed multi-stage SSVEP speller paradigms to improve performance. Sadeghi et al. used commonly occurring characters in the first stage of the interface and infrequently occurring characters in the second stage to improve the efficiency of the speller [67]. To estimate the ITR more accurately, a new definition of it was provided by them. Cao et al. investigated a sliding control paradigm (ITRs of 23.45 bpm), and proved that it outperforms the traditional static protocol on a two-stage SSVEP number speller (ITRs of 19.85 bpm) [68].

### 3.3. One-Stage SSVEP Speller

Compared with the multi-stage SSVEP speller, the one-stage SSVEP speller has the advantage that users need only one command to select their target characters, which often implies higher input efficiency in applications. However, as the one-stage paradigm needs to present all targets at one level, it incurs stringent requirements on the design of the frequency of the stimulus, especially when using a stimulus presentation device with a refresh frequency of 60Hz. Many studies have considered this problem, and one proposed solution is to add a phase difference to stimuli at the same frequency [47]. 

#### 3.3.1. QWERTY Paradigm

Hwang et al. proposed an SSVEP speller that uses a QWERTY-style keyboard with 30 LEDs flickering at different frequencies. As shown in Figure 5a, the frequencies of flicker of the 30 LEDs are all different, and range from 5.0 Hz to 7.9 Hz with a span of 0.1 Hz. This speller achieved an average accuracy of 87.58% and an ITR of 40.72 bpm. In addition, they showed that a difference of frequency of 0.1 Hz presented by the paradigm was useful for the detection of the SSVEP [69].

#### 3.3.2. RC Paradigm

The row column (RC) paradigm is widely used in the P300 speller. Characters in the same row or column flash simultaneously, and the system can determine the character selected by the user through coordinates of the row and column. Yin et al. implemented an SSVEP speller by using the RC normal form, as shown in Figure 5b. In this speller, characters in each row and column use the same stimulus frequency to find the row and column of the target, and the alternating flashing of columns and rows can be used to determine the position of the target. When dynamic optimization was used, this speller achieved a significantly higher practical ITR (PITR) than that achieved by using the fixed optimization approach [70].

#### 3.3.3. Frequency–Phase Hybrid Coding (FPHC) Paradigm

As both frequency and phase have been shown to be identifiable, they can be used to present more identifiable stimuli to the user without increasing the number of stimulus frequencies. Both four (with a span of 0.5π) and five phases (with a span of 0.4π) have been shown to be identifiable in Reference [71], which means that four or five times the original number of target stimuli can be presented by keeping the number of stimulus frequencies constant. The FPHC paradigm can thus reduce the number of stimulus frequencies required by the one-stage SSVEP speller on the premise of maintaining the number of targets.

Nakanishi et al. proposed an SSVEP speller based on an FPHC paradigm. In this speller, eight frequencies and four phases were encoded in a hybrid manner to present 32 stimuli to the user. This study achieved an average 91.35% of accuracy and an average 166.91 of bpm ITR, which was the record at the time [47]. Chen et al. proposed a 40-target speller [71] and compared the performance of two hybrid target-coding strategies on it: (1) mixed frequency and phase coding (using eight frequencies and five phases), and (2) joint frequency and phase coding (using 40 frequencies and four phases), as shown in Figure 5c,d. The results show that mixed frequency and phase coding was superior, with an average accuracy of 89.21% and an ITR of 172.37 bpm.

### 3.4. Specific Paradigms

Some studies have proposed unique SSVEP spellers that do not belong to the above two kinds of spellers. Cao et al. proposed an SSVEP speller with 42 targets in three pages. Each page had 16 flickers arranged in a 4×4 matrix, including 14 flickers to directly select characters and two to select pages. Ideally, users can select the target in up to two steps. The average expense and ITR of this speller were 98.78% and 61.64 bits/min, respectively [72].

The DTU (Technical University of Denmark) speller featured three consistent areas on the screen. The area to the left was a two-stage SSVEP speller with seven flickers. Each corresponded to a group of seven symbols, and was active during the character selection stage. The middle area contained a constantly active flicker to choose the stage and a text box to display the entered characters. The area to the right contained five flickers presenting words suggested from a built-in language model dictionary. They were active when the user switched between the stages of word selection. The average accuracy and ITR of this speller were 90.81% and 21.94 bits/min, respectively [73].

Akce et al. developed an SSVEP speller with adaptive queries [59]. There were 2925 distinct range queries and 20,475 character queries. It queried the given dataset dynamically according to the content entered by the user and the content in the query pool until the user selected the next character. A wider and more accurate spelling prediction is important for improving the efficiency of the speller. This study used a newly defined parameter, named the information gain rate (IGR), to evaluate the performance of the speller. This speller achieved an IGR of 11.93 cpm. 

To enhance the practicability of the SSVEP speller in application scenarios, a BCI speller that can directly output words [74] or sentences [75] has also been designed. Although it is not as flexible as the method that directly selects the character, it has high practical value in specific application scenarios, such as simple question-answering systems and alarm systems that use only a few commands.

### 3.5. High-Frequency Stimuli-Based SSVEP Speller

Compared with the low-frequency stimulus paradigm, the high-frequency paradigm has better and more stable classification performance [76,77,78]. Won et al. proved this by comparing high-frequency stimuli (26–34.7 Hz) with low-frequency stimuli (6–14.9 Hz) in a scene by using the QWERTY paradigm. The difference was obtained because the SSVEP evoked by the high-frequency stimuli was not easily disturbed by waves in the resting state. Chabuda et al. proposed a two-stage SSVEP speller by using high-frequency stimuli. This speller allows the user to select eight frequencies at stimuli in the range 30–39 Hz, and achieved an average accuracy of 89% and an ITR of 36 bpm, which was the highest speed of all high-frequency SSVEP-based BCIs at the time [79]. In a later study on evaluating the performance of different frequency bands in the case of high-frequency stimuli, 35–40 Hz was found to be the frequency band that yielded good performance, with an accuracy of 99.2% and ITR of 67.1 bpm on an SSVEP speller [80].

## 4. SSVEP-Based Hybrid BCI Spellers

The SSVEP can be combined with other triggering methods to form a hybrid speller, such as by using other EEG signals (e.g., the ERP P300), eye-tracking (ET), electrooculogram (EOG), and electromyography (EMG). These hybrid spellers can be divided into three categories. The first consists of spellers that use the SSVEP to promote other triggering methods, the second consists of those that use other triggering methods to improve performance, and the third category consists of spellers that combine the SSVEP and other triggering methods to form new hybrid spellers. We now discuss hybrid spellers as classified by the triggering method used.

### 4.1. SSVEP-P300 Speller

As the SSVEP and the P300 are both based on EEG signals and their areas of detection are independent of each other, a hybrid speller based on the SSVEP and the P300 is feasible and does not require an additional data acquisition device. Although there is competition between them—that is, when the two are triggered at the same time, the intensity of the potential decreases—it does not affect their detection capability [81].

Panicher et al. proposed an approach using the SSVEP for controlling state (CS) detection in the P300 speller [82]. The same method was also implemented in Reference [83]. As the CS has only two states (control and non-control), this method does not require frequency coding, and needs only one frequency stimulus to detect the CS. This method has been shown to improve the performance of the P300 speller.

The SSVEP-P300 speller proposed by Yin et al. is based on the P300 speller by using the RC normal form, where the target is coded along the sub-diagonal to evoke the SSVEP [84]. As shown in Figure 6a,b, this coding method ensures that there is no target flashing at the same frequency inside the same row or column. The target detection includes the detection of coordinates of its row and column as well as that of the frequency of the SSVEP. It improves spelling accuracy, yielding an average accuracy of 93.85% and an ITR of 56.44 bpm.

In subsequent research, Yin et al. proposed an SSVEP-P300 speller based on the subarea/location (SL) paradigm, an RC paradigm-based SSVEP-P300 speller that uses the SSVEP as column stimulus and the P300 as row stimulus, and showed that it improves performance [85]. In addition, two 64-target SSVEP-P300 spellers were proposed in another study by them: the double RC (DRC) and the 4D paradigm [86]. In both spellers, the target changes in color and angle to evoke the P300 and a two-step SSVEP is used. The frequencies in the DRC speller are encoded according to rows and columns while those in the 4D speller are encoded according to the diagonals and sub-diagonals. The 4D speller achieved better performance than the DRC speller in experiments. Both spellers have been shown to be superior to the P300 and SSVEP spellers alone when the same normal form is used.

Xu et al. also proposed an SSVEP-P300 hybrid speller [87] in which a fixed flicker frequency is used as background to evoke the SSVEP. When the target lights up, it evokes a P300 potential. At the same time, the SSVEP signal disappears because the target is not flashing. This process is known as SSVEP blocking (SSVEP-B), and the performance of the hybrid speller based on the SSVEP-B-P300 is superior to that of the original P300 speller. The GUI of the SSVEP-B-P300 speller proposed in Reference [88,89] is shown in Figure 6c. It has four parallel sub-spellers, with flashing at different frequencies as the background. The characters in each sub-speller light up randomly, and induce the P300 and SSVEP-B. The SSVEP evoked by background flicker is used to select the sub-speller, and the SSVEP-B and P300 are used to select characters in the sub-speller. This speller achieved an average accuracy of 87.8% and an ITR of 54 bpm.

Chang et al. proposed an SSVEP-P300 speller that can elicit dual-frequency SSVEP [90]. As shown in Figure 6d, this speller consists of nine panels flickering at different frequencies, each containing four targets. A dual-frequency SSVEP is evoked by the flashing panels and periodically converted characters. This paradigm can improve the performance of the speller by solving the limitation of the harmonic frequency of the SSVEP and the round time of the P300.

Reference [91] reported a hybrid speller based on the P300 speller by using the RC paradigm and adding a peripheral-field SSVEP, and the authors showed that this paradigm has higher accuracy than the P300 speller while not causing more visual fatigue in the user. The LSC-4Q speller proposed in Reference [92] introduced the SSVEP to the P300 speller without directly adding frequency coding to the target to evoke the SSVEP to improve the performance of the P300 speller. Loughnane et al. proposed a gaze-independent hybrid BCI based on the P300, SSVEP, and alpha-band modulation [93]. This paradigm exploits the sensitivity of the SSVEP to covert attention and parieto-occipital alpha band activity.

Rapid serial visual presentation (RSVP) is also used in conjunction with the SSVEP to implement a hybrid speller. Jalipour et al. introduced an SSVEP to solve the problem of reduced classification accuracy due to the evocation of the P300 by non-target stimuli in a triple RSVP paradigm [94]. The results of experiments showed that the introduction of the SSVEP ensures adequate accuracy while improving the ITR. The accuracy of the results was significantly higher than that of a single RSVP and the ITR was significantly higher than that of the triple RSVP.

### 4.2. SSVEP-ET Speller

In addition to EEG signals, other methods of triggering can also be combined with the SSVEP to design hybrid spellers. As a visual triggering method similar to the SSVEP, ET is highly compatible with the SSVEP. ET data processing typically detects the position of the user’s gaze from images captured by a camera to identify the area where the target is located. Early studies used the ET as an auxiliary method to improve the performance of the SSVEP speller [95,96]. Subsequent SSVEP-ET hybrid spellers divided the target into groups selected by using the ET. When a group was selected, the SSVEP was used to select a target from the group. Structurally, this type of SSVEP-ET paradigm is very similar to the multi-stage SSVEP speller.

Reference [95] added a web-camera to the original QWERTY SSVEP speller [69] to record ET, and its paradigm is shown in Figure 7. The ET data will be used for classification aided judgment to select “left” or “right.” The SSVEP is still used to detect the corresponding target according to frequency. When the result of detection of the SSVEP is inconsistent with the position detected by the ET, the speller does not input the given character. The introduction of the ET can prevent spelling errors and reduce deletion operations caused by erroneous inputs, thus improving the accuracy and efficiency of the input. The relevant system achieved an average accuracy of 87.58% and an ITR of 40.72 bpm.

In the SSVEP-ET speller proposed by Mannan et al., 48 targets were divided into eight groups and displayed in a one-stage paradigm. The same frequency set (consisting of six frequencies) was used between groups, and each target within the same group used a different frequency, as shown in Figure 8. The ET was used to identify the group at which the user was gazing and the SSVEP to determine the specific target of the user within the group. The average accuracy and ITR of this speller were 90.35% and 190.73 bpm, respectively [97]. 

Reference [98] provided an opposite approach to the above-mentioned research. Both the SSVEP and the ET were used to select the target group in the primary stage. This was a speller consisting of two stages and 36 targets, where the latter were divided into nine groups. The target groups were arranged on a screen in a 3 × 3 matrix in the primary stage. Each column used the same frequency set, and the flashing frequencies in the three target groups were different, as shown in Figure 9. In the primary stage, the ET was used to select the left, middle, and right columns, and the SSVEP was used to determine the flicker frequency. The results of both were used to judge the target group of the user’s gaze. Once the target group had been selected, the secondary stage was activated. It presented four targets in four directions of the screen, and the ET was used to select the target according to the direction of the user’s gaze.

Some studies have proposed a more flexible and dynamic paradigm [99,100,101] that involves extracting a small region by first detecting fuzzy areas of the user’s gaze. The system then encodes the targets in the extracted region by frequency to evoke the SSVEP and uses it to detect the target in the region at which the user is staring. Yao et al. combined the SSVEP-ET speller with virtual reality (VR) [93]. In a matrix paradigm representing 40 targets, four targets around the gaze point were obtained by detecting eye movement, and a 2 × 2 sub-matrix was highlighted on the interface. The sub-matrix used joint frequency–phase modulation, i.e., the frequency and phase of each target were different, thus presenting four targets to the user. The SSVEP was used to identify targets from the sub-matrix. The average accuracy and ITR of this study were 95.2% and 360.7 bpm, respectively. Reference [100] proposed a dynamic two-stage SSVEP-ET speller; once the ET had recognized the area of the user’s gaze, this area was presented on the screen alone instead of being enhanced (Figure 10).

### 4.3. SSVEP-EOG Speller

In addition to using images to detect eye movement, the EOG can be combined with the SSVEP to design a hybrid speller. Unlike the ET, the EOG does not detect the user’s gaze but their specific eye movements, such as blinking. Saravanakumar et al. proposed a one-stage hybrid speller [102] and a two-stage hybrid speller [103] based on the EOG and the SSVEP. In both studies, the EOG was used to select regions or groups while the SSVEP was used to identify targets. In Reference [102], an SSVEP-EOG speller and an SSVEP-ET speller were compared on a one-stage paradigm and delivered similar performances. In subsequent research, visual feedback was added to the SSVEP-EOG speller, and its performance improved. Reference [103] used nine eye movements that can be recorded by EOG to select the target group as shown in Figure 11, which means that it is feasible to implement a more complex input by adding more frequency codes to the SSVEP-EOG speller.

### 4.4. SSVEP-EMG Speller

A combination of EMG and the SSVEP can be used in the design of a hybrid BCI speller. Lin et al. proposed a hybrid BCI speller based on EMG and the SSVEP [104]. The GUI of this speller is shown in Figure 12, in which 60 targets are divided into four groups using the same frequency set. EMG is used to select the group in which the target is located and the SSVEP is used to select targets within the group according to flicker frequency. This is similar to the multi-stage speller. This study also demonstrated that the hybrid speller proposed is faster and more efficient than EMG or the SSVEP alone.

In another study, Rezeika et al. proposed a hybrid speller based on EMG and the SSVEP [105]. Unlike the one summarized above, this speller uses a one-stage 30-target paradigm. EMG is not used for stage or group selection, but to confirm the results to avoid input errors. This approach improves the control capability of people who have difficulties when using the SSVEP speller.

## 5. Factors Influencing Performance and Visual Fatigue

In addition to the paradigm-based designs of the SSVEP speller, hybrid methods of triggering, and methods of SSVEP detection, other factors influence the performance of the BCI speller when using the SSEVP.

In terms of parameter optimization, individual differences lead to the absence of a fixed combination of parameters to enable every user to achieve the best performance. Therefore, it is necessary to calibrate each subject in the experiment to ensure that they can perform at their highest level, which significantly limits the design and results of experiments on the SSVEP speller [106]. Gembler et al. designed a system that automatically determines user-dependent key parameters to customize SSVEP-based BCI systems. Using this wizard, 61 subjects obtained an average of accuracy of 97.02% and ITR of 21.58 in a three-stage SSVEP speller [107]. In another study, an adaptive time segmentation method obtained an even higher ITR by adaptively selecting thresholds [108]. These two methods can improve the performance and efficiency of the system by reducing the time needed to calibrate users through the SSVEP speller, which can help popularize its application.

In terms of GUI, the currently known infusion factors include the number of targets and methods of feedback. The impact of the number of targets on the performance of the speller is definitive. When the speller presents more targets, the performance of the speller degrades. In the method proposed in Reference [109], the ITR cache reached its peak when the number of targets presented at the same time in the paradigm was 15. Different feedback methods have also been shown to influence the accuracy and spelling time of spellers. This influence is not identical for different users, and thus there is no superior solution [26]. However, good user feedback can help improve the overall performance of the speller and increase its speed of input without affecting accuracy [27].

In the experimental environment, the distance between the user and the device used for stimulus presentation affects the performance of the speller, where the performance deteriorates with increasing distance. One to two meters is a suitable distance [110]. Reference [111] showed that background music affects the user’s input accuracy and speed.

The age of the subject has a significant impact on performance. In two studies on the effects of age on performance, the ITRs of young and old people were 27.36 bpm and 16.19 bpm (about 169%) in Reference [112], and 27.18 bpm and 14.42 bpm (about 188%) in Reference [113]. On the same system, young people outperformed older subjects. In addition, practice by subjects has been shown to be effective for improving performance [114].

Considering that it is a problem that needs to be addressed by the visual BCI speller, the literature has researched reducing the visual fatigue experienced by users. In the design of the stimuli of the paradigm, high-frequency stimuli are thought to cause less fatigue than low-frequency stimuli. At the same time, because the duty cycle has no prominent effect on the user’s fatigue level, it can be designed at 50% to increase efficiency [78]. In another study, the central-field SSVEP (cSSVEP) and the peripheral -field SSVEP (pSSVEP) were compared in the context of reducing visual fatigue while ensuring adequate decoding accuracy [81].

## 6. Trends, Challenges, Prospective Directions, and Suggestions

In the above, we reviewed past studies on SSVEP-based spellers and hybrid spellers. Table 1 and Table 2 present a summary. Our analysis has revealed four main means of performance optimization for the SSVEP speller: improving the classification algorithm, adding a spelling prediction function, designing better paradigms, and adding new triggering methods. We also identified some shortcomings in current research in the area.

### 6.1. Development Trends

The aforementioned four methods of optimization can be summarized with regard to the overall trend of development in BCI spellers that use SSEVP as below.

First, novel methods of SSVEP detection are constantly being proposed. From the early power spectrum density analysis, represented by the FFT [31], to the subsequent and widely used spatial filtering method, including the well-known CCA [35] and improved algorithms [41,42,43,44,45,46,47,48] as well as the advanced TRCA [50], methods of SSVEP detection are being updated quickly, as mentioned in Section 2. Deep learning-based methods are also being used [29,52] but not widely because they require a significant amount of training data.

Second, word prediction has improved the performance of spellers [18,24,59,61,66,73]. Adding a word prediction function to the SSVEP speller can enable users to directly select words when a spelling prompt presents it, instead of having to enter every letter of it. This is important for improving the efficiency of the speller, especially in application scenarios where users need to input a large amount of text. With the development of natural language processing (NLP) technology, spelling prediction has become easier to implement. In addition, in earlier studies, to reduce the time required to correct spelling errors, researchers integrated error-related potentials (ErrPs) into P300 spellers [115,116,117,118]. ErrPs can be evoked by unexpected responses which contain user’s mistake and error of the speller, and researchers use it to automatically detect these spelling errors. However, in practical applications, limited by the difficulty of collecting sufficient training data, ErrP is not widely used. Some researchers considered that the ErrP corrective mechanisms can be replaced by other methods, and proposed the corresponding alternative method [119]. At present, the word prediction can also alleviate this problem by reducing the occurrence of errors.

Third, many paradigms have been proposed. In past studies, more than 20 paradigms have been proposed one after another, and can be mainly divided into two kinds: multi-stage paradigms [64,65,66,67,68] and one-stage paradigms [69,70,71]. It is easy to conclude that the multi-stage speller is easier to design and implement because of its less stringent requirement on the number of frequencies of the stimulus, but the ITR is relatively slow. The one-stage speller allows users to select targets in one step but the design of the frequencies of the stimulus is more complicated, and often requires more frequencies or frequency–phase hybrid coding. In addition to these two kinds, many paradigms have been proposed, including those not mentioned in Section 3. It is conceivable that new paradigms will continue to be proposed unless a prevalent paradigm proves to be optimal.

Finally, combining the SSVEP with other triggering methods has attracted significant research attention. The SSVEP is easy to be evoked in independent positions (occipital regions), and is compatible with most stimuli. Many SSVEP-based hybrid spellers have been proposed, and have delivered better performance than any single triggering method-based speller. Section 4 has provided several triggering methods that have been combined with the SSVEP to form a hybrid speller, including the visual ERP P300 [84,85,86,87,88,89], ET [96,97,98,99,100,101], EOG [102,103], and EMG [104,105]. 

### 6.2. Outstanding Challenges

#### 6.2.1. Lack of Comparison of Paradigms

Although the stimulus paradigm is commonly considered to have a significant impact on the performance of BCI spellers, no study to date has compared the available paradigms to identify the one that delivers the best performance using the control variable method. As shown in Table 1, when the same methods of SSVEP detection are used, the results obtained are inconsistent owing to the different paradigms used, and thus the influence of the paradigm on the performance of the SSVEP speller cannot be ignored. However, the feedback method used, number of targets, calibration method used, and the relevant parameters also affect its performance. Thus, past work cannot be used to determine the best paradigm. 

#### 6.2.2. Lack of Research on System-Independent Factors Influencing Results

In addition to the design of the speller, system-independent factors may also affect the results in experiments. Factors that can affect the experimental results include the distance between subjects and the screen [110], background music [111], and the age of the subjects [112,113]. However, the relevant results lack quantitative analyses. At the same time, some pertinent factors have been ignored in previous studies.

#### 6.2.3. Lack of Research on Visual Fatigue

In addition to performance, the degree of fatigue on the user when using the speller should be considered from the perspective of applications. Visual fatigue leads to poor user experience, and a worse user experience often means worse application prospects. Although some studies have given methods to evaluate user fatigue [8,120,121], no unified standard has been developed to date, and research on this issue is inadequate. The challenge faced by such research is that it is very difficult to quantitatively analyze the degree of fatigue, which often depends on the subjective feelings of the subjects. 

### 6.3. Directions for Future Research

Based on the above trends of research and challenges, the authors proposed the following ideas for future research in the area. First, deep learning is an advanced method of SSVEP detection but has not been widely used in the context of the SSVEP speller. The use of deep learning as a method of SSVEP detection has been shown to improve the classification performance of the system, and PodNet has been shown to yield better performance than the FBCCA while using fewer channels [52]. Deep learning requires a large amount of data that are not yet publicly available. Moreover, the methods of collection and paradigms of these datasets are not the same, which limits their application. However, Reference [54,55] proposed methods that use transfer learning across devices and montages to enable the use of data collected by using different methods for the same research. This is important for strengthening the application of deep learning to SSVEP detection. In addition, methods besides deep learning should be examined to optimize prevalent techniques.

Second, applying NLP methods can improve the performance of spellers. Akce et al. used 2925 distinct range queries and 20,475 character queries in their study [59]. The use of NLP methods in their work far exceeded that in previous studies. This marked the introduction of advanced NLP methods to the SSVEP speller as a means of further improving its performance. In addition, individual optimization through a statistical analysis of users’ personal language habits, performance, and user experience of the speller can be improved. 

Third, novel paradigms for the SSVEP-based speller and hybrid speller are expected. In addition to comparative studies on the SSVEP speller paradigm, more modes of EEG triggering, such as the auditory P300 and MI, and non-EEG triggering methods can be added to the design of hybrid spellers in future research.

### 6.4. Suggestions for Improvements

To improve the performance of prevalent SSVEP spellers, researchers should develop optimization strategies based on the above methods. Such strategies should consist of: i.Adding known effective and compatible auxiliary triggering methods;ii.Adding or improving the word prediction function;iii.Using advanced methods of SSVEP detection, such as the TRCA;iv.Adjusting the experimental environment to avoid system-independent interference, andv.Optimizing methods of calibration to select better parameters for subjects.

As there is no consensus on an appropriate paradigm, we do not have any suggestions on this issue. Some factors are known to affect the results of experiments but other factors may also be revealed in future research. Studies on system-independent factors are valuable for a quantitative comparison of the performance of the SSVEP speller. At the same time, if conditions permit, experiments should be undertaken on MND patients as subjects. This is important for testing the application prospects of SSVEP spellers for those who need them. In addition, more attention should be paid to the level of visual fatigue induced in the user because this is an unavoidable problem in applications.

## 7. Conclusions

In this review, we summarized BCI spellers that use the SSVEP from the viewpoint of the stimulus paradigm and performance. The stimulus paradigm includes the procedure used for target selection, layout of the targets, manner of encoding of the stimuli (i.e., frequency and phase), and their combination with other triggering methods. These options within paradigms influence the performance of the BCI speller. 

The one-stage SSVEP speller has higher input efficiency than the multi-stage speller, and should be considered if the device has the capacity of presenting more frequencies of the given stimulus. Both the frequency and the phase can be used to encode the targets of the stimulus. A high-frequency stimulus paradigm delivers a better and more stable classification performance than a low-frequency one. The number of stimulus targets can range from 30 to over 60. The SSVEP can be combined with other triggering methods, including the ERP P300, ET, EOG, and EMG to construct hybrid BCI spellers that deliver better performance.

In addition, the optimization of stimulus paradigms, individual calibration and user feedback improve the performance of BCI spellers based on the SSVEP. The distance between the users and the stimulus presentation device, as well as the age of users, influences the experimental results.

Research on [120] a comparison of paradigms, system-independent influential factors, and visual fatigue is lacking. The performance of the BCI speller when used by MND patients and SSVEP-illiterate users has not been well studied. Future research in the area should focus on deep learning for SSVEP detection, combinations with NLP methods, novel SSVEP stimulus paradigms, other methods of triggering, and an examination of visual fatigue in the user.

## Figures and Tables

**Figure 1 brainsci-11-00450-f001:**
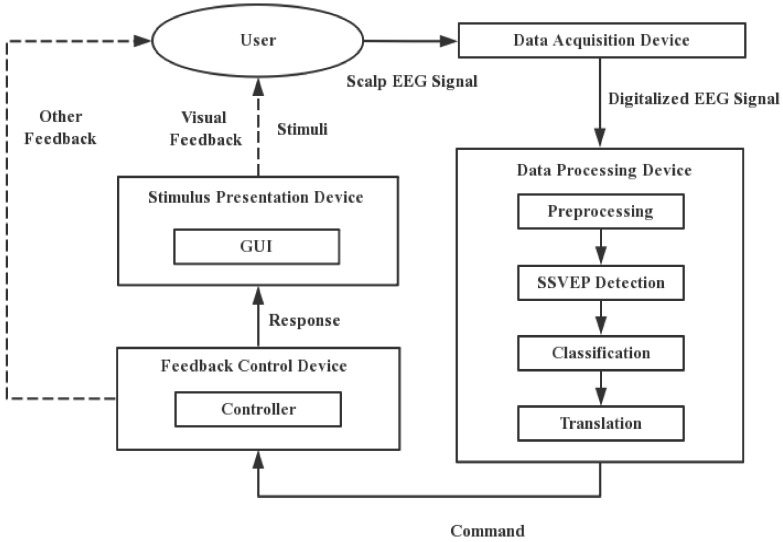
The architecture and process of a steady-state visual evoked potential (SSVEP)-based brain–computer interface (BCI) speller. The solid lines in the figure represent the data flow within the system and the dashed lines represent the interaction between the system and the user. EEG indicates electroencephalograph; GUI indicates graphical user interface.

**Figure 2 brainsci-11-00450-f002:**
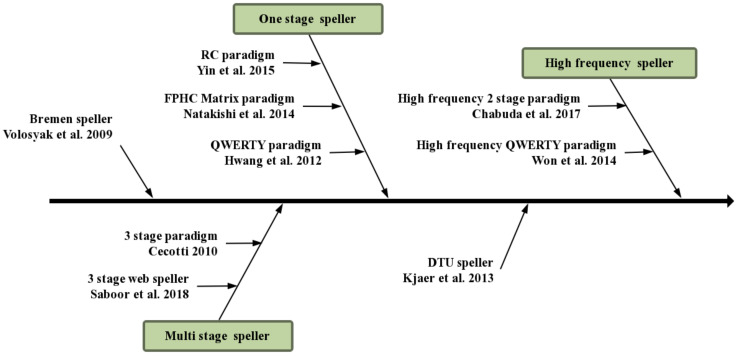
Subdivisions of the steady-state visual evoked potential (SSVEP) based BCI speller according to paradigms. RC: row and column; FPHC: frequency–phase hybrid coding.

**Figure 3 brainsci-11-00450-f003:**
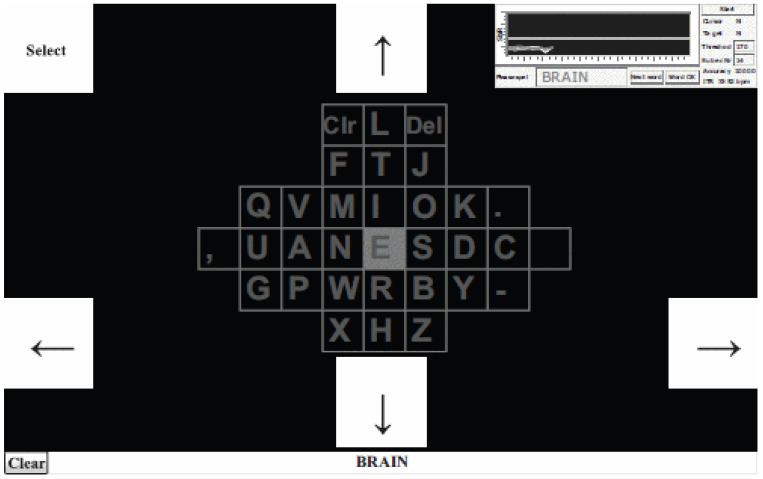
Graphical user interface (GUI) of the Bremen speller. © 2009 IEEE. Reprinted, with permission, from [60].

**Figure 4 brainsci-11-00450-f004:**
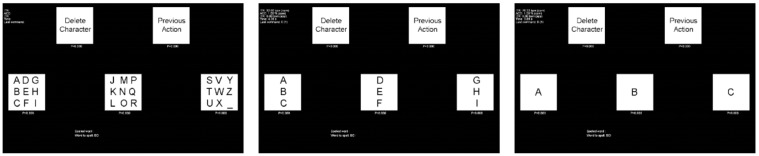
The graphical user interface (GUI) of one multi-stage steady-state visual evoked potential (SSVEP) speller as an example. © 2010 IEEE. Reprinted, with permission, from [64].

**Figure 5 brainsci-11-00450-f005:**
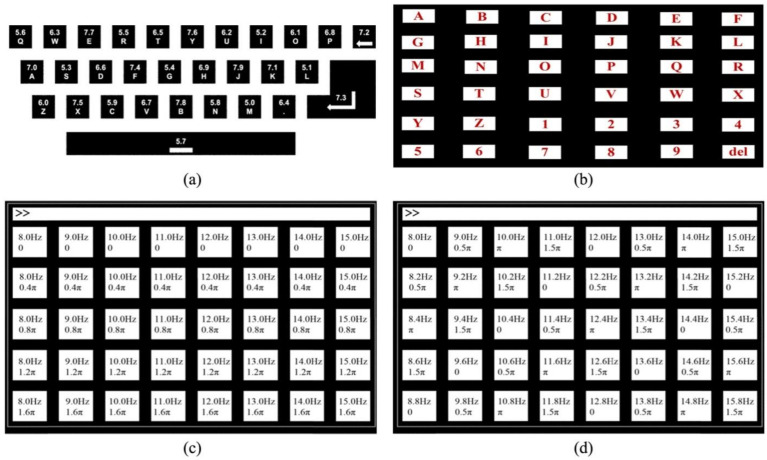
The paradigms of one-stage steady-state visual evoked potential (SSVEP) spellers. (**a**) The frequency arrangement of the QWERTY paradigm. Reprinted from [69], with permission from Elsevier. (**b**) Graphical user interface (GUI) of the SSVEP speller using the RC paradigm. © 2015 IEEE. Reprinted, with permission, from [70]. (**c**) Mixed frequency and phase coding. © 2014 IEEE. Reprinted, with permission, from [71]. (**d**) Joint frequency and phase coding. © 2014 IEEE. Reprinted, with permission, from [71].

**Figure 6 brainsci-11-00450-f006:**
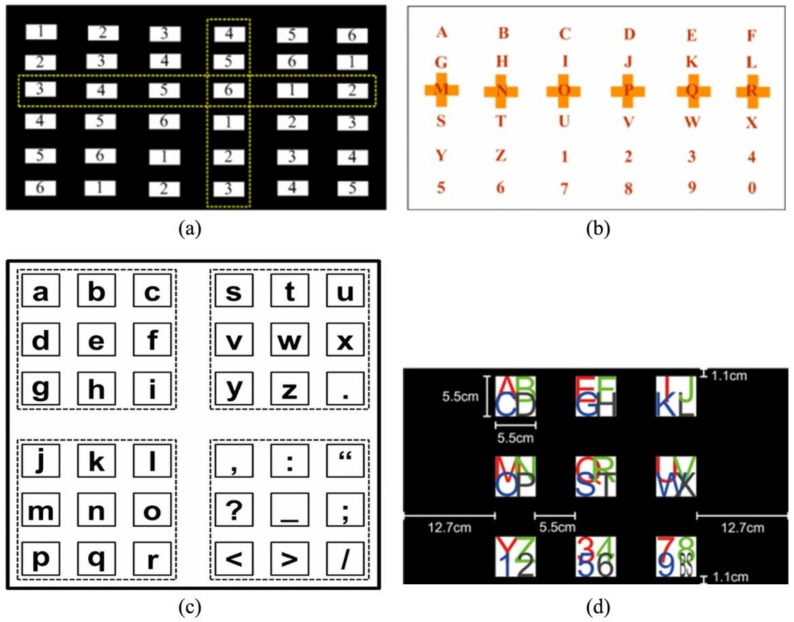
Three examples of the SSVEP (steady-state visual evoked potential) -P300 speller. (**a**) The frequencies’ array where different numbers represent different frequencies and subareas surrounded by the dashed lines flash at the same time to evoke the P300 potential. (**b**) GUI of the SSVEP-P300 speller proposed in Reference [84]. © IOP publishing. Reproduced with permission. All right reserved by [84]. (**c**). The graphical user interface (GUI) of the SSVEP-B-P300 speller (subareas surrounded by the dashed lines flash at the same frequencies to evoke the SSVEP potential. Only one of the targets of each sub-speller is enhanced to evoke a P300 potential and an SSVEP-B potential). © IOP publishing. Reproduced with permission. All right reserved by [88,89]. (**d**). GUI of the SSVEP-P300 proposed by Chang et al. [90].

**Figure 7 brainsci-11-00450-f007:**
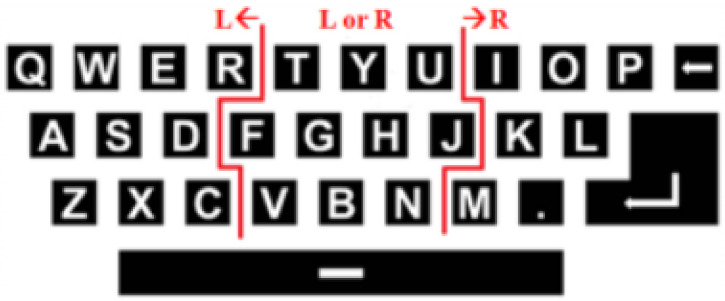
Graphical user interface (GUI) of the steady-state visual evoked potential (SSVEP) -eye-tracking (ET) speller using the QWERTY paradigm (the area to the right of the right red line is recognized as “right.” The area to the left of the left red line is recognized as “left.” The area between the lines is recognized as “left” or “right.”). © 2013 IEEE. Reprinted, with permission, from [95].

**Figure 8 brainsci-11-00450-f008:**
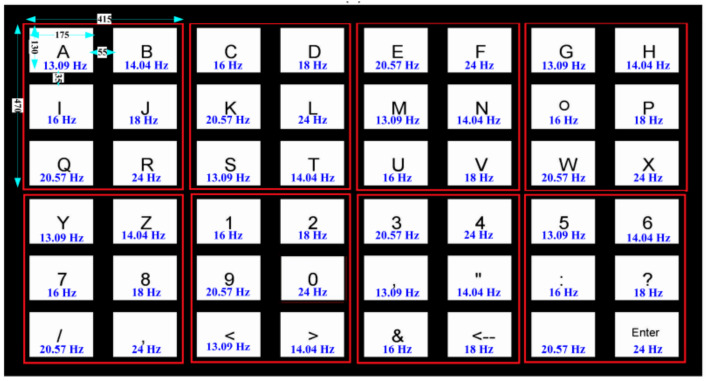
The arrangements of the sub-areas and frequencies of the the steady-state visual evoked potential (SSVEP) -eye-tracking (ET) speller proposed by Mannan et al. [97] (the eight sub-areas surrounded by red lines are distinguished by the ET).

**Figure 9 brainsci-11-00450-f009:**
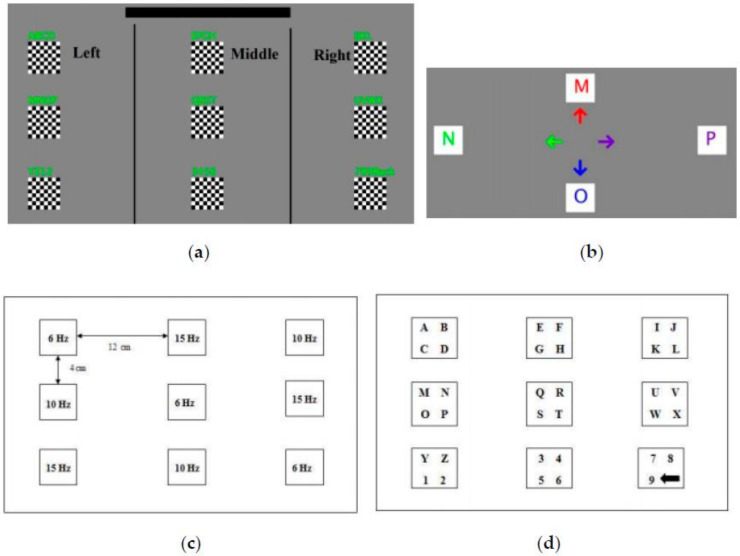
The steady-state visual evoked potential (SSVEP) -eye-tracking (ET) speller proposed by Saravanakumar and Reddy [98]. (**a**,**b**) Primary stage of the paradigm. (**c**,**d**) Secondary Scheme 2018. IEEE. Reprinted, with permission, from [98].

**Figure 10 brainsci-11-00450-f010:**
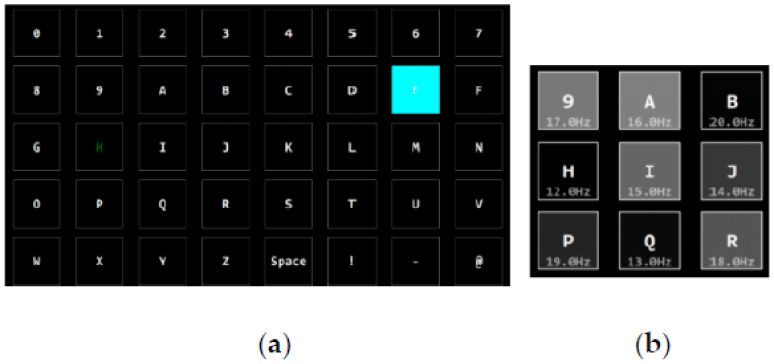
One steady-state visual evoked potential (SSVEP) -eye-tracking (ET) speller as an example. (**a**) Graphical user interface (GUI) of the primary stage. (**b**) GUI of the secondary stage of the SSVEP-ET speller. © 2019 IEEE. Reprinted, with permission, from [100].

**Figure 11 brainsci-11-00450-f011:**
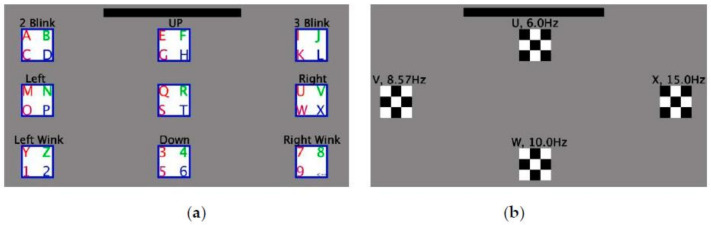
An example of the steady-state visual evoked potential (SSVEP) -electrooculogram (EOG) speller. (**a**) The paradigm of the primary stage. (**b**) The arrangement of frequencies of the secondary stage. Reprinted from [69,103], with permission from Elsevier.

**Figure 12 brainsci-11-00450-f012:**
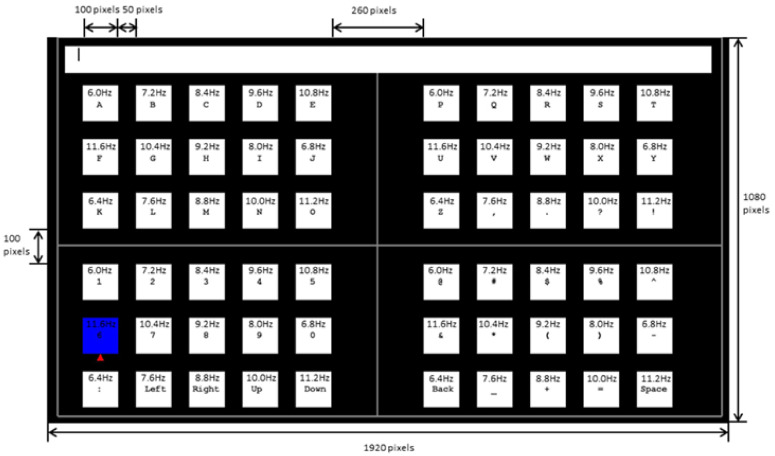
The paradigm and the arrangement of frequencies of the steady-state visual evoked potential (SSVEP) -electromyography (EMG) speller. © IOP publishing. Reproduced with permission. All right reserved by [104].

**Table 1 brainsci-11-00450-t001:** Summary of SSVEP spellers. ITR: information transfer rate; PITR: practical information transfer rate; MEC: minimum energy combination; CCA: canonical correlation analysis; PSD: power spectrum density; IGR: information gain rate.

Reference	SSVEP Detection Method	Paradigm	Number of Commands/Targets	Frequencies of Stimuli (Hz)	Subjects	Accuracy (%)	ITR (bpm)
Volosyak et al., 2009	Principal Component Analysis, (PCA)	Bremen speller	5 commands 32 targets	6.67, 7.5, 8.57, 10, 12	37 29 healthy	92.84	17.4
Cecotti, 2010	PCA	Multi-stage speller (3 stages)	5 commands 27 targets	6.67, 7.5, 8.57, 7.06, 8	8 8 healthy	92.25	37.62
Saboor et al., 2018	MEC	Multi-stage speller (3 stages)	4 commands 27 targets	6, 7.5, 8, 8.5, 9, 9.5, 10	10 10 healthy	94.50	12.74
Nguyen et al., 2018	PodNet	Multi-stage speller (3 stages)	5 commands 58 targets	6.67, 7.5, 8.57, 10, 12	8 8 healthy	97.37	48.99
Hwang et al., 2012	Threshold	One-stage speller QWERTY	30 targets	5–7.9 (span 0.1)	10 10 healthy	87.58	40.72
Yin et al., 2015	CCA-RV (canonical correlation analysis with reducing variation)	One-stage speller RC	36 targets	8.18, 8.97, 9.98, 11.23, 12.85, 14.99	11 11 healthy	72.28	41.08 (PITR)
Nakanishi et al., 2014	CCA with SSVEP training data	One-stage speller FPHC	32 targets	8–15 (span 1) 4 phases 0 0.5π, 1.5π	13 13 healthy	91.35	166.91
Chen et al., 2014	Extended CCA	One-stage speller FPHC	40 targets	8–15 (span 1) 5 phases 0 0.4π, 0.8π, 1.2π, 1.6π	6 6 healthy	89.21	172.37
				8–15.8 (span 0.2) 4 phases 0 0.5π, 1.5π		88.83	170.94
Cao et al., 2011	CCA	One-stage 3 pages	16 commands 42 targets	8–15.5 (span 0.5)	4 4 healthy	98.78	61.64
Kjaer et al., 2013	Threshold	DTU speller (2 stages + one stage)	8 commands 49 targets + 6 commands 5 targets	6, 6.5, 7, 7.5, 8.2, 9.3, 10, 11	9 9 healthy	90.81	21.94
Akce et al., 2015	PSD	Query-based speller	5 commands 29 targets	6.67, 7.5, 8.57, 10,12	11 11 healthy	98.5	11.93 IGR (cpm)
Chabuda et al., 2017	Time domain comb filter	Multi-stage speller (2 stages)	8 commands 36 targets	30–39 (span 1.0)	15 15 healthy	89	36

**Table 2 brainsci-11-00450-t002:** Summary of hybrid spellers using the SSVEP. SL: subarea/location.

Reference	Paradigm	Triggering	Number of Targets	Subjects	Accuracy (%)	ITR (bpm)
Panicker et al., 2011	One-stage RC	SSVEP +P300	36	10 10 healthy	88	19.05
Yin et al., 2013	One-stage RC	SSVEP +P300	36	12 12 healthy	93.85	56.44
Yin et al., 2015	One-stage DRC	SSVEP +P300	64	13 13 healthy	91.33	47.14
	One-stage 4D				95.18	50.14
Xu et al., 2014	One-stage SL	SSVEP-B +P300	36	11 11 healthy	87.8	54
Chang et al., 2015	One-stage SL	Dual frequency SSVEP +P300	36	10 10 healthy	93	31.8
Hwang et al., 2013	One-stage QWERTY	SSVEP + ET	30	10 10 healthy	87.58	40.72
Mannan et al., 2020	One-stage matrix	SSVEP +ET	48	20 20 healthy	90.35	190.73
Saravanakumar et al., 2018	Multi-stage (2 stages)	SSVEP +ET	36	10 10healthy	90.46	65.98
Yao et al., 2018	One-stage matrix (VR) (dynamic)	SSVEP +ET	40	3 3 healthy	95.2	360.7
Lin et al., 2019	Multi-stage (2 stages) (dynamic)	SSVEP +ET	40	5 5 healthy	92.1	180.8
Saravanakumar et al., 2019	One-stage matrix	SSVEP + EOG	36	10 10 healthy	98.33	69.21
Saravanakumar et al., 2020	Multi-stage (2 stages)	SSVEP (stage 2) + EOG (stage 1)	36	10 10 healthy	94.16	70.99
Lin et al., 2016	Multi-stage (2 stages)	SSVEP + EMG	60	10 10 healthy	85.8	90.9
Rezeika et al., 2018	One-stage matrix	SSVEP + EMG	30	8 8 healthy	93.75	31.05

## Data Availability

No new data were created or analyzed in this study. Data sharing is not applicable to this article.

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
