# Peer review of "Brain–Computer Interface Speller Based on Steady-State Visual Evoked Potential: A Review Focusing on the Stimulus Paradigm and Performance"

_brainsci, 2021, doi:10.3390/brainsci11040450_

Round 1

Reviewer 1 Report

This work includes a comprehensive review of SSVEP-based BCI spellers. In general, the manuscript is structured fairly well and covers the main concepts regarding this type of BCI. Nevertheless, I have various comments that I feel need to be addressed before this work can be accepted for publication. Please see my comments below:

- The authors do well at introducing the need for BCI technology and how P300 and SSVEP signals have been used to address this need, however, they also overlook sensorimotor rhythm (SMR)-based BCIs. For completeness, the authors should briefly mention SMR BCIs and describe some of the prominent accomplishments achieved with this signal. For example, an SMR BCI was recently used for the continuous control of a robotic arm, a feat only previously achieved using intracortically acquired signals. Please see the reference below:

BJ Edelman et al. (2019) Noninvasive neuroimaging enhances continuous neural tracking for robotic device control. Science Robotics 4(31):eaaw6844

Similarly, while this review focuses on BCI spellers, the authors should also acknowledge that SSVEP signals have been used to control practical devices such as wheelchairs. This should also be briefly mentioned in the introduction, with support by references such as the one below:

Y Li et al. (2013) A hybrid BCI system combining P300 and SSVEP and its application to wheelchair control. IEEE TBME 60(11):3156-66

- It is generally accepted that a BCI is comprised of four components rather than three, as the authors describe in section 2.1. While the three components that the authors mention are correct, they do no include the concept of user feedback (often visual in the case of an SSVEP BCI). This component is necessary to close the loop between the computer/machine and the user so that he/she can modulate neural signals and complete the desired task. This section should be revised to account for this fourth piece of a BCI system. Please see the reference below regarding this topic:

G Schalk et al. (2004). BCI2000: a general-purpose brain-computer interface (BCI) system. IEEE TBME 51(6):1034-43

- The authors provide comprehensive descriptions of detection algorithms used for SSVEP-based BCIs, however, include quite little discussion regarding error potentials and how they have been used in these systems. While error potentials are particularly relevant to P300-based BCIs, a concise description of this concept would be beneficial to the hybrid SSVEP-P300 content.

- In general, 16 figures is a lot for an article, even for a large review such as this one. I suggest that the authors try to combine figures with multiple panels to condense the content. 

- In sections 6.2.3 and 6.3, the authors include discussion points regarding MND patients that seem to be a bit random. Even though this is a prime target population for BCI systems, patient testing seems to be out of the scope of the current work and therefore this content appears out of place. I would recommend that the authors remove or cut down the emphasis on this patient group in these sections, and the manuscript in general. 

Reviewer 2 Report

In this review authors undertake the problem of SSVEP phenomena applied to speller interfaces. The speller interface is one of many possible BCIs and as authors conclude the one-stage SSVEP speller has higher input efficiency than the multi-stage speller. That is why some other triggering methods like ERP P300, ET, EOG, and EMG could be combined with SSVEP in order to construct some hybrid spellers. Authors perfectly present the current state of the art and do not miss any important papers in the field according to my knowledge. 

I suggest authors their own implementation of Sych hybrid speller which could be reported in our journal.

Author Response

The authors would like to thank the reviewer for the positive comments. Once we finish our own Sych hybrid speller, we will submit the work to Brain Sciences for considering to be published.

Round 2

Reviewer 1 Report

I thank the authors for their efforts in revising the manuscript. I believe all of my comments have been addressed and think the work is now suitable for publication. Nevertheless, please check for minor language/grammar errors before final publication.